# Breast Cancer—Epidemiology, Classification, Pathogenesis and Treatment (Review of Literature)

**DOI:** 10.3390/cancers14102569

**Published:** 2022-05-23

**Authors:** Beata Smolarz, Anna Zadrożna Nowak, Hanna Romanowicz

**Affiliations:** 1Laboratory of Cancer Genetics, Department of Pathology, Polish Mother’s Memorial Hospital Research Institute, Rzgowska 281/289, 93-338 Lodz, Poland; hanna-romanowicz@wp.pl; 2Department of Chemotherapy, Medical University of Lodz, Copernicus Memorial Hospital, 93-513 Lodz, Poland; anna.m.zadrozna@gmail.com

**Keywords:** breast cancer, risk factors, pathomorphology, therapy

## Abstract

**Simple Summary:**

Breast cancer is the most-commonly diagnosed malignant tumor in women in the world, as well as the first cause of death from malignant tumors. The incidence of breast cancer is constantly increasing in all regions of the world. For this reason, despite the progress in its detection and treatment, which translates into improved mortality rates, it seems necessary to look for new therapeutic methods, predictive and prognostic factors. The article presents a review of the literature on breast carcinoma - a disease affecting women in the world.

**Abstract:**

Breast cancer is the most-commonly diagnosed malignant tumor in women in the world, as well as the first cause of death from malignant tumors. The incidence of breast cancer is constantly increasing in all regions of the world. For this reason, despite the progress in its detection and treatment, which translates into improved mortality rates, it seems necessary to look for new therapeutic methods, and predictive and prognostic factors. Treatment strategies vary depending on the molecular subtype. Breast cancer treatment is multidisciplinary; it includes approaches to locoregional therapy (surgery and radiation therapy) and systemic therapy. Systemic therapies include hormone therapy for hormone-positive disease, chemotherapy, anti-HER2 therapy for HER2-positive disease, and quite recently, immunotherapy. Triple negative breast cancer is responsible for more than 15–20% of all breast cancers. It is of particular research interest as it presents a therapeutic challenge, mainly due to its low response to treatment and its highly invasive nature. Future therapeutic concepts for breast cancer aim to individualize therapy and de-escalate and escalate treatment based on cancer biology and early response to therapy. The article presents a review of the literature on breast carcinoma—a disease affecting women in the world.

## 1. Epidemiology

Breast cancer is the most common malignant tumor in women in the world. Breast cancer patients account for as much as 36% of oncological patients. An estimated 2.089 million women were diagnosed with breast cancer in 2018 [1,2]. The incidence of this malignant tumor is increasing in all regions of the world, but the highest incidence occurs in industrialized countries. Almost half of the cases on a global scale are in developed countries [2,3]. This trend is mainly due to the so-called Western lifestyle, associated with a poor diet, nicotinism, excessive stress and little physical activity [3]. In the case of breast cancer, mammography has become recognized as screening. The greatest value of mammography is observed in the group of women aged 50–69 years [1,3]. Classical mammography is characterized by 75–95% sensitivity and specificity at the level of 80–95% [4]. For women with suspected hereditary breast cancer, magnetic resonance mammography is used as a screening test. If a suspicious lesion is found in mammography, an ultrasound examination is performed and, if necessary, a thick needle biopsy along with a histopathological examination of the tumor.

In 2018, there were 234,087 cases of breast cancer in the United States (crude rate: 85/105), 55,439 in the United Kingdom (crude rate: 94/105), 56,162 in France (crude rate: 99/105), 71,888 in Germany (crude rate: 85.4/105) and 66,101 in Japan (crude rate: 58/105) [2]. The highest incidence rate in the world is found in Belgium (crude rate: 113/105), and among the continents—in Australia (crude rate: 94/105) [2]. In Poland, breast cancer is also the most-commonly diagnosed malignant tumor in women. There is a steady increase in cases (1990, 8000 new cases; 2018, 20,203 new cases) [2]. The average incidence rate in Europe is 84/105 [2]. The lowest incidence occurs in the countries of Southeast Asia and Africa, where the standardized incidence rate does not exceed 25/105 [2]. The lowest incidence rates in 2018 were recorded in Bhutan (crude rate: 5/105) and the Republic of The Gambia (crude rate: 6.5/105) [2]. Despite the greater effectiveness of initial diagnostics or the rapid development of pharmacotherapy in recent years, breast cancer is the first cause of death from malignant tumors in women in the world. In 2018, 626,679 people died from breast cancer. Unlike morbidity, the highest mortality from this malignant tumor is recorded in developing countries [2] (Fiji, crude rate 36/105, highest rate; Somalia, crude rate 29/105; Ethiopia, crude rate 23/105; Egypt, crude rate 21/105; Indonesia, crude rate 17/105; Papua New Guinea, crude rate 25/105) [2], in which as much as 60% of all deaths from breast cancer occur. This trend is mainly related to the lack of screening, which is less than in developed countries, the availability of diagnostics and modern methods of treatment [5]. In contrast, the standardized death crude rate in Belgium 16.3/105, in the United States 13/105, and in Japan 9.3/105 [2]. The number of breast cancer cases in Poland is much lower than in EU countries (in 2013, the standardized incidence rate for Polish—51.8, for the EU 106.6) [6]. The incidence of adult premenopausal women (20–49 years) has almost doubled over the past 30 years. Unfortunately, Polish women are still not very sensitive to prevention. They neglect their breasts and underestimate the importance of regular check-ups. Compared to other European countries, Polish women have a low incidence of preventative care—in the Netherlands, 80% of women report free mammogram prevention programs, in England 71%, and in Poland only 44% [6]. The percentage of 5-year survival due to breast cancer in Poland is 78.5%, differing significantly from, for example, the result of 90% achieved in the United States [7].

## 2. Risk Factors

The unambiguous cause of carcinogenesis has not yet been established, but several risk factors conducive to the development of breast cancer are known. One of the most important, as also indicated by the epidemiological data described above, are the gender, age, and degree of economic development of a given country. No less important are hormonal factors, mainly related to the time of exposure to estrogens, procreative factors, including the number of children born, the age of birth of the first child, or breastfeeding. Great importance in the development of breast cancer is attributed to genetic factors, the use of hormone replacement therapy, improper diet, and the resulting obesity. Among the significant risk factors for the development of breast cancer, hormonal contraception, alcohol consumption and exposure to ionizing radiation at a young age are also mentioned. Risk factors for breast cancer are presented in Table 1.

### 2.1. Sex

The vast majority of cases of breast cancer, reaching 99%, occur in women. Only 1% of cases of this malignant tumor affect men, for which the standardized incidence rate in Poland is 0.4/105. No more than 100 cases are reported each year [6]. However, the incidence of breast cancer in men, like that in women, shows a steady upward trend, which is most likely associated with obesity and longer survival [9].

### 2.2. Age

Age is one of the most important risk factors for breast cancer. The global increase in the incidence of breast cancer is observed in all age groups and is highest in women under 50 years of age [9]. Although this malignant tumor is rare in this age group, it remains a significant clinical and social problem, due to its worse course—numerous studies indicate that breast cancer in young women is characterized by greater histological malignancy, marginal expression of steroid receptors, frequent overexpression of the HER-2 receptor or occurs as a molecular biological subtype “basal-like” (“triple negative”) [10]. Furthermore, the incidence of breast cancer in premenopausal women is increasing—within 30 years it has increased almost 2-fold [6].

### 2.3. Degree of Economic Development

As mentioned in the paragraph on epidemiology, the incidence of breast cancer and mortality from this malignant tumor is related to the economic development of a country. This relationship has been documented in many studies [3,11,12].

The incidence of breast cancer is increasing worldwide due to the continuous growth of the population and the ageing of the population [11]. The highest incidence rates are recorded in developed countries [3,11,12]. This phenomenon results from the so-called “Western lifestyle” described above. At the same time, it seems that soon the trend of high morbidity will also occur in developing countries. In these countries, along with economic development, access to public health care becomes easier, prevention and screening programs are introduced (which increases detection), maternal, infant and child mortality decreases [12]. On the other hand, the importance of factors conducive to the development of breast cancer is growing, such as late first birth, low number of babies born, use of hormone replacement therapy, obesity, lack of physical activity, or improper diet [11,12]. Currently, however, lower middle- and low-income countries are dominated by higher breast cancer mortality rates than in developed countries, despite lower incidence [3,5,11,12]. This trend is associated with frequent diagnosis of cancer at an advanced stage, which results from the lack of resources for the effective implementation of primary prevention programs, diagnostic tests (primarily mammography), and finally modern methods of treatment [5,11,12].

Approximately 645,000 cases of premenopausal breast cancer and 1.4 million cases of postmenopausal breast cancer were diagnosed worldwide in 2018, with more than 130,000 and 490,000 deaths in each menopausal group, respectively. Proportionally, countries with a low UNDP Human Development Index (HDI) faced a higher burden of premenopausal breast cancer for both new cases and deaths compared to higher-income countries [13]. Countries with very high HDI had the highest incidence of premenopausal and postmenopausal breast cancer (30.6 and 253.6 cases per 100,000, respectively), while countries with low and medium HDI had the highest premenopausal and postmenopausal mortality rates (5 and 53.3 deaths per 100,000, respectively). By studying trends in breast cancer, they noted significantly increasing age-standardized incidence rates (ASIRs) for premenopausal breast cancer in 20 of 44 populations and significantly increasing ASIRs for postmenopausal breast cancer in 24 of the 44 populations. Growth only in premenopausal age occurred mainly in high-income countries, while the increase in the burden of postmenopausal breast cancer was most noticeable in transition countries [13].

### 2.4. Hormonal Status

Factors related to a woman’s hormonal status seem to have a huge impact on the risk of developing breast cancer. The results of many studies indicate that the risk of developing breast cancer increases in proportion to the time of exposure to estrogen, which prolongs early menarche, late menopause, the age of birth of the first child and the number of children born [9,11,12,13,14].

Brinton et al. showed that the first menstruation that occurred at or after the age of 15 was associated with a 23% reduction in the risk of breast cancer compared to the first menstruation before the age of 12 (early menarche) [15]. Currently, it is believed that this reduction is about 30%. In turn, the Collaborative Group on Hormonal Factors in Breast Cancer published in 2012 in The Lancet Oncology the results of a meta-analysis, according to which the relative risk of developing breast cancer increased by 5% with each year of early menarche initiation [16].

In addition, it was found that early first menstruation was associated with a higher risk of developing breast cancer compared to late menopause—each year of late menopause increased the relative risk by 2.9%, with late menopause being believed to be for, achieved after the age of 54, increases the risk of breast cancer twice compared to the menopause achieved before the age of 45 [1,16].

The meta-analysis also showed that women who had not reached menopause had a higher risk of developing breast cancer compared to postmenopausal women of the same age. In this group of analyzed patients, the effect of the Body Mass Index on the risk of developing the disease was noticed—in premenopausal patients, obesity reduced this risk, and in postmenopausal patients it increased. This metanalysis also found that early menarche was associated with a higher risk of developing lobular breast cancer, as was late menopause [16]. Late menopause also predisposed to developing steroid-expressed breast cancer [16].

Other reproductive factors with an effect on breast cancer risk confirmed in numerous studies include the age at which the first child was born, the number of babies born and breastfeeding [14].

Studies indicate an increased risk of breast cancer in transgender women compared to cisgender men and a lower risk in transgender men compared to cisgender women [17]. In transgender women, the risk of breast cancer increases during a relatively short period of hormone treatment, and the characteristics of breast cancer are more like a female pattern. The results of the study suggest that guidelines for breast cancer screening for cisgender people are sufficient for transgender people using hormone treatment [17].

Recent studies indicate that the increased risk of breast cancer is associated with long-term use of estrogen-only therapy and combined estrogen-progestogen therapy [18]. The combination treatment associated with the least increase in risk is estradiol-dydrogesterone. Research suggests a lower increased risk of breast cancer associated with long-term hormone replacement therapy (HTR) use and a more noticeable decrease in risk after discontinuation of treatment [18].

### 2.5. Reproductive and Hormonal Risk Factors in Breast Cancer Patients

Estrogens play an important role in the pathogenesis of the development of breast cancer [19]. Breast cancer is considered a hormone-dependent tumor in which elevated estrogen levels and longer exposure to this hormone are associated with an increased risk of its development [19].

This is confirmed by epidemiological studies that increased exposure to endogenous and exogenous estrogens increases the risk of developing breast cancer [20].

In all postmenopausal women, high serum estrogen levels are associated with an increased risk of breast cancer. Both hormonal factors and reproductive factors are indisputably influencing the increase in the risk of breast cancer [20]. The duration of exposure to estrogen and the effect of pregnancy determined by parameters such as the age of the first menstrual bleeding, the age of the first pregnancy (especially exposure in women who gave birth to the first child after the age of 30), childlessness, or the age of onset of menopause change the individual risk of breast cancer [21].

Early onset of menstruation (12 years) and late termination (50 years) increases the risk twice compared to women who started menstruation late (15 years) and ended it early (40 years) [22]. 

Also, childlessness and the late age of the first pregnancy (over 30 years of age) are factors associated with prolonged exposure to estrogens [23]. 

Nulliparous women and those who became pregnant for the first time after the age of 30 have an increased risk of getting sick 2–5 times more. Spontaneous and artificial miscarriages (incomplete pregnancies) do not confer a protective effect as do full pregnancies, but they may increase the risk due to the lack of protective effect of progesterone in the second phase of pregnancy [24]. The effect of exogenous estrogen on breast cancer is a controversial issue and continues to be subjected to numerous studies. 

The use of HRT is a significant risk factor for breast cancer. The first information on the adverse effects of HRT on the risk of developing breast cancer appeared in the nineteen nineties. In 1997, the Collaborative Group on Hormonal Factors in Breast Cancer published in The Lancet the results of a meta-analysis of 51 studies evaluating the relationship between HRT intake and breast cancer. This meta-analysis found that each year of HRT use increased the risk of breast cancer by 2.7% [25]. In 2019, the same society republished another meta-analysis in The Lancet, this time of 58 prospective studies evaluating the relationship between the type of HRT and the risk of developing breast cancer. This meta-analysis showed that HRT containing estrogens and progestogens increased the risk of breast cancer to the greatest extent, especially when progestogens were taken daily [25]. The use of HRT even for a short time (1–4 years) was also associated with an increased risk of breast cancer [25]. The risk of developing the disease was mainly related to breast cancer with the expression of steroid receptors [25]. The risk of developing the disease was slightly reduced if HRT was used after the age of 60 [24]. This risk was also lower for obese women, especially if they took HRT containing only estrogens [26].

Two-component HRT, used for 5 years from the age of 50, was associated with a 2% increase in the risk of breast cancer over 20 years in patients aged 50–69 (from 6.3% to 8.3%)—it is estimated that 1 in 50 women would develop cancer [26]. Similarly, the use of HRT with estrogens and progestogens taken intermittently increased the risk of breast cancer from 6.3% to 7.7% (1 in 70 women would get sick). In turn, single-component HRT (only with estrogens) used for 5 years was associated with the lowest increase in the risk of breast cancer in 20 years—from 6.3% to 6.8% (1 in 200 women would get sick) [26].

The relationship between hormonal contraceptive use and breast cancer risk has been demonstrated in two important papers—a reanalysis of 54 epidemiological studies by the Collaborative Group on Hormonal Factors in Breast Cancer published in The Lancet in 1996, and a prospective cohort study by Mørch et al. presented in the NEJM in 2017 [27,28]. Both studies found that long-term use of hormonal contraception adversely affects the relative risk of breast cancer. This risk was estimated at 1.20 (Danish study) and 1.24 (CGoHFiBC reanalysis). It was higher the longer the subjects took hormonal contraception (1.09 for hormonal contraception used for less than a year vs. 1.38 for women taking contraception for more than 10 years) [27,28].

In addition, this cohort study showed that the relative risk of developing breast cancer was elevated for at least 5 years after the end of hormonal contraception in women who took it for a long time (≥5 years). This trend was not noticed in women who used hormonal contraception for a short time (less than 5 years) [28].

The relative risk of developing breast cancer was also increased regardless of the type of contraception taken [27].

### 2.6. Genetic Factors, Family Occurrence

Only a small group of breast cancer cases (5–10%) are genetic. The best-known genetic mutations associated with this cancer include mutations in the *BRCA1* and *BRCA2* genes [29].

The *BRCA1* gene, located on chromosome 17, is a suppressor gene that encodes nuclear protein, responsible for maintaining genome stability. Together with the products of other suppressor genes, signal transduction genes and DNA damage detection, this protein co-creates a protein complex that binds to RNA polymerase II and interacts with histone deacetylase, thus affecting the processes of transcription, DNA repair or recombination. The *BRCA1* protein, together with the *BRCA2* gene product, which is also a suppressor gene located on chromosome 13, is particularly involved in the repair of double DNA strand breaks by homologous recombination [30].

The presence of mutations in these genes occurs only in 3–5% of breast cancer patients. However, due to the high penetration of *BRCA1/BRCA2* genes, these patients should be included in the prophylactic program. Carriers of the *BRCA1/BRCA2* mutation are estimated to have a 10-fold higher risk of developing breast cancer [27]. The presence of *BRCA1/BRCA2* gene mutations is associated with a cumulative risk of breast cancer at age of 70 of more than 60%, and the probability of developing this malignant tumor throughout life varies in the range of 41–90%. Mutations in the *BRCA1* gene are associated with triple-negative cancer and in the *BRCA2* gene for estrogen receptor-expressed breast cancer [31,32,33].

Other suppressor genes whose high-penetration mutations predispose to breast cancer are the TP53 (Li-Fraumeni syndrome) and PTEN (Cowden syndrome) genes. The cumulative risk of developing breast cancer at age 70 for women with Li-Fraumeni syndrome is 54%. In patients with Cowden’s syndrome, the risk of developing breast cancer throughout life is in the range of 25–50%. However, both genetic syndromes are very rare [34,35].

Mutations in the *ATM*, *BRIP1*, *CHEK2* and *PALB2* genes show a moderate predisposition to breast cancer. Carriers of these mutations have a 2–3 times higher risk of developing this malignant tumor [35].

It is believed that <10% of breast cancers are genetically determined [36]. More than 90% of breast cancers, on the other hand, are formed because of sporadic somatic mutations. The risk of developing breast cancer increases twice in women whose closest relative (mother, sister) has been treated for the malignant tumor in question and by three to six times if the two closest relatives have been treated [1]. This risk decreased the older the relative was at the time of diagnosis of cancer [1].

### 2.7. Mild Breast Changes

Another factor that increases the risk of developing breast cancer is the presence of benign changes in the mammary glands. Some benign lesions—benign neoplasms, e.g., atypical ductal hyperplasia (ADH) or atypical lobular hyperplasia (ALH), which increase the risk by four or five times, and proliferative (proliferative) lesions without atypia (e.g., stellar scar or fibrotic adenoma) increasing the risk up to two times. The Hartmann et al. cohort study assessed the risk of breast cancer in patients with different types of benign lesions [33]. The relative risk of developing breast cancer for the entire study cohort was 1.56 (95% CI, 1.45–1.68) [37]. This risk was elevated for 25 years after the biopsy. For women with benign lesions without proliferation, the relative risk of developing breast cancer was 1.27 (95% CI, 1.15–1.41). In the presence of mild proliferating lesions, but without atypia, it was equal to 1.88 (95% CI, 1.66–2.12). The highest relative risk of developing breast cancer was for women with the presence of benign proliferating lesions with atypia (atypical ductal hyperplasia, atypical lobular hyperplasia, or both), amounting to as high as 4.24 (95% CI, 3.26–5.41). It was also found that the earlier benign changes were diagnosed (<55 years of age), the greater the risk of developing the malignant tumor in question [37].

In addition, it is believed that in women with atypical hyperplasia, whose first-degree relatives were treated for breast cancer, the risk of developing this malignant tumor is as much as nine times greater [38].

### 2.8. Ionizing Radiation

A recognized factor in the development of breast cancer is early exposure to ionizing radiation. In 2007, John et al. published an analysis of data from the Breast Cancer Family Registry assessing the relationship between exposure to ionizing radiation used in diagnosis and treatment and the risk of developing breast cancer [39]. This analysis showed that an increased risk of breast cancer was in women who had received radiation therapy in the past as part of cancer treatment and in women who underwent a control chest X-ray during treatment for tuberculosis and pneumonia [39]. The risk of developing breast cancer was highest in young patients whose exposure to ionizing radiation was multiple and in patients who had exposure at a very young age [39].

In the study Moskowitz et al. published in 2014, the risk of developing breast cancer was assessed depending on the dose and field of radiotherapy in women who were exposed to the chest area due to cancer (leukemia, Hodgkin’s or non-Hodgkin lymphoma, Wilms’ tumor, neuroblastoma, soft tissue sarcoma, bone malignant tumor, tumor of the central nervous system) before the age of 21 [39]. This study indicated that the highest risk of developing breast cancer was in patients who were treated with radiation therapy at lower doses (14 Gy) but for a large chest area (whole lung field), consequently covering a larger area of breast tissue [38]. The risk of developing breast cancer in women who received high-dose radiotherapy (30–40 Gy) for a smaller chest area (Mantle field) was comparable or lower (mediastinal field) but elevated compared to women who had not been irradiated in the past [39]. The risk of developing breast cancer was lower if the radiation field included the ovaries [39]. It was also shown that the cumulative risk of developing breast cancer at the age of 50 was 30%, with the highest (35%) in patients treated for Hodgkin lymphoma [40].

In a systematic review by Henderson et al., it was also found that the highest risk of developing breast cancer occurred in patients treated in the past for Hodgkin lymphoma. It was noted that the analyzed studies mostly concerned such patients [41].

In 2005, a paper by Travis et al. was published assessing only the relationship between breast cancer and radiotherapy received in the chest area for Hodgkin lymphoma. The study showed that the cumulative absolute risk of developing breast cancer increased with the patient’s age, sometimes after the diagnosis of cancer and the dose of irradiation [42].

It was mentioned above that performing a control chest X-ray during the treatment of tuberculosis and pneumonia increased the risk of developing breast cancer. As for other diagnostic methods, it is also believed that mammography performed in young women significantly increases the risk of breast cancer [1].

Ionizing radiation (IR) increases the risk of breast cancer, especially in women and when exposed at a younger age, and the evidence generally supports the linear dose-response relationship [43]. Ionizing radiation directly and indirectly causes DNA damage and increases the production of reactive oxygen and nitrogen species (RONS). The RONS lead to DNA damage and epigenetic changes leading to mutation and genomic instability. Proliferation of RONS enhances the effects of DNA damage and mutations leading to breast cancer. Separately, damage to reactive oxygen and nitrogen species and DNA also increases inflammation. Inflammation contributes to direct and indirect effects (effects in cells unattainable directly by IR) through positive feedback to RONS and DNA damage, and separately increases proliferation of breast cancer through pro-carcinogenic effects on cells and tissues. For example, changes in gene expression alter inflammatory mediators, resulting in improved cancer cell survival and growth and a more hospitable tissue environment [43]. All these events overlap at multiple points with events characteristic of “basic” breast cancer induction, including hormone-dependent proliferation, oxidative activity, and DNA damage. These overlays make the breasts particularly susceptible to ionizing radiation and confirm that these biological activities are important characteristics of carcinogens [43].

### 2.9. Alcohol Consumption

Numerous studies indicate a relationship between alcohol consumption and an increased risk of breast cancer [44,45,46,47,48]. This dependence results from several mechanisms —alcohol contributes to the increase in the concentration of estrogens in the blood by inhibiting their metabolism in the liver and by intensifying the conversion of androgens to estrogens. In addition, it has an inhibitory effect on the immune system, or DNA repair processes, may intensify cellular proliferation and migration. Finally, the metabolites of alcohol themselves are carcinogenic compounds [49]. It is estimated that for every consumption of 10 g of pure alcohol per day, there is an increase in the risk of breast cancer by 9% [1].

### 2.10. Diet

The influence of the type of diet used on the development of the cancer process has been the subject of numerous studies. The correlation between a low-varied diet, rich in saturated fats, including those of animal origin, and the risk of developing mainly colorectal cancer seems undeniable [50]. On the other hand, studies assessing the relationship between diet and the risk of breast cancer are not entirely consistent. Dandamudi et al. reviewed systematic studies published between 2013 and 2017. Ten of the seventeen publications evaluated looked at the impact of a so-called “unhealthy” diet on breast cancer risk. The basic products of the diet in question included: sweetened soft drinks, processed fruit juices, red and processed meats, hardened fats, saturated fats, salted products (chips, chips, peanuts), refined grains, sweetened products (sweets, desserts) [50]. In most, but not all, of the studies analyzed, a significant relationship was found between excessive consumption of the above-mentioned products and an increase in the risk of developing breast cancer. This relationship primarily concerned excessive consumption of red and processed meat, saturated fats, and sodium [51].

This systematic review also showed that a diet rich in vegetables, fruits, fish, legumes, oils, and vegetable oils reduces the risk of breast cancer [51].

Research suggests that nutrition affects the prognosis of breast cancer. Nevertheless, the level of evidence on the results is still insufficient to make recommendations. A healthy and balanced diet should be encouraged to reduce mortality in the world [52].

A healthy diet characterized by a high intake of unrefined grains, vegetables, fruits, nuts and olive oil, and a moderate/low intake of saturated fatty acids and red meat may improve overall survival after a diagnosis of breast cancer. Breast cancer patients undergoing chemotherapy and/or radiation therapy experience various symptoms that worsen patients’ quality of life. Studies on nutritional interventions during breast cancer treatment have shown that nutritional counseling and supplementation with certain dietary components, such as eicosapentaenoic (EPA) and docosahexaenoic (DHA) acids, may be useful in reducing drug-induced side effects as well as increasing therapeutic efficacy. Therefore, nutritional intervention in patients with BC can be considered an integral part of a multimodal therapeutic approach [53]. 

The link between breast cancer and diet is known to be complex, multifactorial, and nonlinear. Classical epidemiological studies on nutrition have shown conflicting results, showing little correlation between diet and breast cancer risk (except alcohol) [54]. It can be speculated that this may be due to the complexity of breast cancer, which is a multifaceted, highly heterogeneous disorder. Histological classifications, and more recently also molecular ones, have contributed to the formation of a rather complex picture.

Nutrigenomics and related disciplines can support advances in knowledge in this field by shedding light on the molecular basis of breast cancer formation and paving the way for personalized therapies.

### 2.11. Obesity

One of the risk factors for developing breast cancer, confirmed in many studies, is obesity. Jiralerspong and Goodwin compiled a pooled analysis of numerous publications evaluating the relationship between obesity and breast cancer prevalence in premenopausal and postmenopausal women. This analysis found that both overweight and obesity increased the risk of developing breast cancer, particularly steroid-receptor-expressed breast cancer, in postmenopausal women who did not use hormone replacement therapy [55,56,57].

Unlike postmenopausal patients, being overweight or obese in premenopausal women reduces the risk of developing hormone-dependent breast cancer. The authors of the analysis point out, however, that literature data indicate a relationship between obesity in premenopausal patients and the risk of developing triple-negative breast cancer [55,58].

This analysis also found that physical inactivity (combined with obesity) increases the risk of developing breast cancer regardless of menopausal status. Furthermore, according to the results of numerous studies, overweight and obesity are associated with a worse prognosis of breast cancer patients before and after menopause [55,59,60]. According to the authors, worse survival may be influenced by a greater stage of cancer at the time of diagnosis, as well as a more aggressive course of breast cancer in obese patients [51]. Obesity promotes the process of cancer through several mechanisms. Overdeveloped adipose tissue is a source of numerous cytokines, chemokines, endocrine factors, in particular proangiogenic and promitogenic leptin, which affects the immune environment of the described tissue [61]. There is a concentration of cells of the immune system of a pro-inflammatory nature, additionally secreting inflammatory cytokines. Excessive development of adipose tissue promotes the surrounding hypoxia, which leads to an increase in the secretion of leptin and VEGF factor, while inhibits the synthesis of antiangiogenic and antimitogenic adiponectin [62]. The NF-κB (nuclear factor kappa-light-chain-enhancer of activated B cells) pathway is responsible for the development and maintenance of the inflammatory process within excessive adipose tissue, which through pro-inflammatory cytokines has an inhibitory effect on the process of apoptosis, and at a later stage promotes the proliferation of breast cancer cells, cancer invasion, angiogenesis, and metastasis [63,64].

Adipose tissue is also the main source of sex hormones in postmenopausal women. In this tissue, estrogens are formed in the process of aromatization of adrenal androgens. The accumulation of pro-inflammatory cytokines in overgrown adipose tissue, activation of the NF-κB pathway within adipose tissue, or dying adipocytes stimulate the activity of the aromatase complex, which in turn leads to excessive estrogen synthesis and promotes the development of breast cancer [62].

Furthermore, the metabolic syndrome that accompanies obesity is associated with insulin resistance, hyperinsulinemia, increased synthesis of insulin-like growth factor 1 (IGF-1). Studies have shown that insulin resistance and hyperinsulinemia are associated with poorer survival of breast cancer patients [65]. Breast cancer cells also often overexpress the IGF-1 receptor, making this factor considered their potential mitogen [61].

Obesity is a recognized risk factor for breast cancer and the development of relapses, even if patients are properly treated [66]. Obese women are less likely to undergo breast reconstruction than women of normal weight, and those who have undergone surgery experience more surgical complications. In obese women, systemic chemotherapy and hormone therapy are less effective. Obese women are at greater risk of local recurrence than women of normal weight. The effectiveness of cancer treatment is significantly lower in obese women who survive breast cancer [66].

Given the multidimensional effect of overgrown adipose tissue on the development of breast cancer, the campaign against obesity should form the basis for primary prevention of the malignant tumor in question.

### 2.12. Nicotinism

Research reports on the impact of chronic nicotinism on the increased risk of breast cancer are contradictory. However, a study by Jones et al. published in 2017 showed that smoking, especially at the beginning of early peripubertal age or adolescence, was associated with a moderate but statistically significant increase in the risk of developing breast cancer. The relative risk of breast cancer was higher with a positive family history [67]. Nicotine promotes breast cancer metastasis by stimulating N2 neutrophils and generating a pre-metastatic niche in the lung [68]. Chemoresistance effects of nicotine were demonstrated in breast cancer cells. These findings demonstrated the harmful effects of nicotine following metastasis of cancer, owing to the chemoresistance produced through uninterrupted smoking, which may impact the effectiveness of treatment [69]. 

## 3. Pathomorphology

The basis for the diagnosis of breast cancer remains standard pathomorphological diagnostics [70]. The result of histopathological examination should include not only the histological type of the tumor, its degree of histological malignancy, the degree of advancement according to the TNM classification, information on the completeness of the procedure, or infiltration by cancer cells of peritugal vessels, but also the expression of steroid receptors—estrogen and progesterone, HER-2 receptor, and cellular proliferation index Ki67 [71]. 

A reliable assessment of all the above parameters is possible thanks to the examination of material taken by means of a coarse needle biopsy or intra- and postoperative material [72]. The examination of the material obtained by fine needle biopsy does not allow to distinguish between infiltrating and pre-invasive cancer, as well as to assess the state of HER-2. The correct protocol of histopathological examination, considering the biological subtype of the tumor, determines the determination of recognized predictive and prognostic factors, and consequently the selection of appropriate, individual treatment for each patient.

A common classification of breast cancer is the WHO classification [73], for which in 2019, the 5th edition was published. This described cancers, both benign and malignant, of epithelial, mesenchymal, fibroepithelial origin, neuroendocrine neoplasms, breast wart and nipple areola tumors, in addition, breast lymphomas and metastatic changes in the mammary gland. 

A simplified classification of epithelial precursor and invasive lesions is presented in Table 2.

The most common form of infiltrating cancer is cancer without a special type (NST), formerly called wired (70–80%) [1]. It is characterized by a large diversity in terms of cancer cell morphology and the presence of tubular or glandular structures. The second most common invasive breast cancer is lobular carcinoma (10%) [1]. This form of cancer is characterized by a small diversity of cancer cells, very frequent expression of steroid receptors and extremely rare overexpression of the Her-2 receptor [1].

The degree of histological malignancy (G, grade) was introduced due to the significant diversity of biological characteristics of breast cancers within the same histological type in the absence of characteristic morphological features. The classification used to correctly assess the degree of histological malignancy is, recommended by the WHO, the Bloom–Richardson–Scarff classification in Elston–Ellis modification (Table 3).

Originally, the assessment of the degree of histological malignancy concerned only invasive cancer without a special type (NST). Currently, it refers to any infiltrating cancer, excluding medullary and microinvasive cancer. In the case of heterogeneous cancer weaving, the fields with the highest degree of malignancy should be noted [76].

The current VIII edition of the TNM classification was published by the AJCC (American Joint Committee on Cancer) in 2018. According to this classification, histopathological examination should assess the size of the primary tumor (Tumor), the condition of the axillary lymph nodes (Nodes) and the presence of distant metastasis (Metastasis) (Table 4). In the case of the N trait, the location and number of lymph nodes taken should be described, but it must not be less more than 10 nodes, as well as the number of lymph nodes affected by metastases, including micrometastases and isolated cancer cells. Correct assessment of all elements of the TNM classification makes it possible to determine the stage of cancer, which is the most important prognostic factor [68]. In countries where it is not possible to present a prognostic stage of advancement, containing the state of ER, PR and HER-2 receptors, its anatomical version should be used.

## 4. Prognostic and Predictive Factors

### 4.1. TNM

As mentioned earlier, the stage of breast cancer is the most important prognostic factor. According to SEER data, 98.9% of patients with localized disease, 85.7% with regional advancement, and only 28.1% of patients with distant metastases will survive [7].

In addition, all the individual features of the TNM classification have a prognostic significance.

One of the most important prognostic factors is the condition of the lymph nodes (N). According to SEER data, the 5-year overall survival (OS) is 92% for patients with unoccupied regional lymph nodes, 81% with 1–3 lymph nodes occupied, and 57% when metastases were found in four or more lymph nodes. The presence of micrometastases and isolated cancer cells in regional lymph nodes is also of unfavorable prognostic importance [76,77].

The dimension of the primary tumor is also an important prognostic factor. SEER data indicate that 99% of women with a disease confined to the mammary gland and a tumor smaller than 1 cm, 89% with a tumor measuring 1–3 cm and 86% with a tumor of 3–5 cm will survive 5 years [77]. In addition, a tumor with an originally large size predisposes to the involvement of regional lymph nodes. 

The current feature of T4 according to the TNM classification, i.e., invasion of the skin or chest wall, is also associated with a worse prognosis.

### 4.2. Degree of Histological Malignancy

Of slightly less prognostic significance are the histological type and the degree of histological malignancy. Less common cancers, such as tubular, papillary, and medullary, have a better prognosis with a 10% risk of recurrence with prolonged follow-up [78].

Determining the prognosis in the case of frequent cancers, infiltrating NST cancer and lobular cancer, facilitated the introduction of the degree of histological malignancy. Studies have shown that unfavorable prognostic significance is associated with low tumor differentiation (G3). However, it has not been clearly established what impact moderate differentiation has on the prognosis (G2) [79].

### 4.3. Hormonal Receptors

The expression of steroid receptors—estrogenic and progesterone—is particularly important due to the favorable value of both prognostic and predictive value for hormonal treatment. This expression is assessed by immunohistochemical method (IHC) in tissue material fixed in buffered formalin and embedded in paraffin. If tissue material cannot be obtained, the expression of the receptors is assessed in the cytological material fixed in alcohol. The tissue material should come from the infiltrating component of the primary tumor, prior to systemic treatment. Due to the frequent phenomenon of hormonal profile change in metastatic tumors, it is recommended to reassess the expression of steroid receptors in metastatic material.

The scale used to determine the expression of hormone receptors is the Allrad scale, according to which the percentage of stained nuclei of cancer cells (PS 0–5) and the strength of coloration (IS 0–1) should be assessed. The sum of both parameters is the total value of TS (TS = PS + IS 0–8). In practice, however, as justified by the recommendations of the International Breast Cancer Conference of St. Gallen, only the percentage of colored cell nuclei is considered. Any reaction in the ≥1% of cancer cells is considered positive [80,81,82].

In every patient with current steroid receptors, hormone therapy should be used, regardless of age, condition of regional lymph nodes or additional indications for chemotherapy. The efficacy of complementary treatment with tamoxifen and aromatase inhibitors in hormone-sensitive patients has been demonstrated in numerous randomized controlled trials. In turn, the first reports on the prognostic value of primarily the estrogen receptor were published in the second half of the twentieth century [83,84,85,86,87,88,89]. Steroid receptor expression is associated with better prognosis and lower sensitivity to chemotherapy.

### 4.4. HER-2 Receptor 

The prognostic and predictive value for targeted treatment is also the overexpression of the HER-2 receptor or amplification of the HER-2 gene. The HER-2 state should be determined in the histological material. The assessment of the HER-2 status in the cytological material is of lower value because the staining reaction used in the determination of the receptor occurs in the cell membrane, which is easily damaged during a fine needle aspiration biopsy.

Determining the HER-2 status requires the use of two methods—immunopathological at each diagnosis of infiltrating cancer (Table 5) and the method of in situ hybridization in immunohistochemically borderline cases (about 15–20% of cases). About 10% of ambiguous cases show amplification of the *HER-2* gene after in situ hybridization (FISH or CISH), which is interpreted as a positive state. The in situ hybridization method involves counting a copy of the *HER-2* gene (single probe) or a copy of the *HER-2* gene and the number of centromeres of chromosome 17 (double probe). The test result is the average number of copies of the *HER-2* gene per cell or the ratio of the number of copies of the *HER-2* gene to the number of centromeres. Cases without amplification of the *HER-2* gene are treated as negative.

The HER-2 receptor belongs to the family of four ERBB receptors. The first of these—the EGFR receptor (ERBB1), i.e., the epidermal growth factor receptor with tyrosine kinase properties, is a target for many molecularly targeted drugs. Its ligand is epidermal growth factor (EGF) and TGF-α. The HER-2 receptor (ERBB2), the second in the ERBB receptor family, does not have a specific ligand. Its role is to enhance signal transduction by heterodimerization with other ERBB receptors. Heterodimer with ERBB3 receptor is the strongest signal transducing complex. The presence of overexpression of the HER2 receptor or amplification of its gene is an unfavorable prognostic factor, and the introduction of drugs that block the HER-2 receptor, i.e., trastuzumab, T-DM1, pertuzumab, lapatinib, significantly improved the prognosis of patients. In the meta-analysis of phase III studies, min. HERA studies have shown that the addition of trastuzumab, a monoclonal antibody directed against the HER-2 receptor, to adjuvant chemotherapy is associated with a 40% reduction in the relative risk of recurrence and a relative risk of death of 34% compared to left chemotherapy alone [89].

Studies have shown that the improvement in prognosis also applies to patients treated palliatively. The best example of studies confirming the effectiveness of adjuvant trastuzumab therapy in patients with early HER2-positive breast cancer are 4 international randomized trials-HERA, NSABP-B31, NCCTG-N98 and BCIRG 00, of which the HERA study became a registration study in the above indication [90,91,92,93,94]. One of the first studies to assess the importance of trastuzumab in the first line of treatment for patients with advanced breast cancer was Slamon et al. [89]. The study showed that adding trastuzumab to chemotherapy (CHT) in the first line of treatment significantly improved prognosis.

Inhibition of HER2 in breast cancer with HER2 amplification is clinically effective, as demonstrated by the effectiveness of HER kinase inhibitors and HER2 antibody treatment. Although resistance to HER2 inhibition is common in the case of metastasis, specific programs that follow HER2 resistance have not been established. In the work of Smith et al. [95], through genomic profiling of 733 breast cancers with HER2 amplification, enrichment of somatic changes that promote MEK/ERK signaling in metastatic tumors with reduced progression-free survival after anti-HER2 therapy was identified. These mutations, including NF1 loss and ERBB2 activating mutations, are sufficient to mediate resistance to FDA-approved HER2 kinase inhibitors, including tucatinib and neratinib. Moreover, resistant cancers lose their dependence on AKT, undergoing dramatic sensitization to MEK/ERK inhibition. Mechanically, this driver path switch is the result of the activation of MEK-dependent CDK2 kinase. These results define the genetic activation of MAPK as a recurrent mechanism of resistance to anti-HER2 therapy that can be effectively combated with MEK/ERK inhibitors.

Although rare, *HER2* mutations appear as important molecular changes that need to be identified, for example, in patients with metastasis, tumors with *HER2* mutations may respond to specific tyrosine kinase inhibitors. *HER2* mutation may also be a mechanism of resistance to anti-HER2 therapeutic compounds.

### 4.5. Proliferation Rate Ki67

The Ki67 protein, used in the evaluation of the cellular proliferation index, is a nuclear protein present in all phases of cell division, except the resting phase of G0, and therefore in all actively proliferating cells. The Ki67 protein is identified by immunohistochemical method. The percentage of colored testicles of cancer cells is the value of the cell proliferation index Ki67. However, the positive reaction criterion has not been fully established. It is assumed that 20% is the limit of low and high proliferation.

Currently, the assessment of the cellular proliferation index Ki67 is an essential element of the pathomorphological study, allowing to determine the final luminal subtype of cancer (A or B) and the degree of histological malignancy (G).

The high proliferation index has an unfavorable prognostic significance not only as a component of histological malignancy, but also as an independent prognostic factor [93].

### 4.6. Polygenic Prognostic Factors

The development of molecular biology and genetics allowed for the separation of many new prognostic factors (mainly genes), and the introduction of new technologies to create tools for their determination. These tools are multi-gene predictive tests, currently used to estimate the risk of relapse in individual patients and the benefits of the proposed treatments. In practice, these tests are primarily used to qualify patients with early luminal cancer for adjuvant chemotherapy, in addition to standard hormone therapy. The most well-known tests are Oncotype DX and Mammaprint, of which only Oncotype DX was included in the VIII edition of the TNM classification [28].

### 4.7. pCR

One of the prognostic factors widely commented on recently is the complete pathological response (pCR) obtained through induction chemotherapy. Evaluated in several studies, some contradictory, it has been meta-analyzed by Spring et al. and published in Clinical Cancer Research in 2020. This meta-analysis showed that the achievement of pCR as a result of preoperative systemic treatment was associated with an increase in event-free survival (HR = 0.31; 95% PI, 0.24–0.39), especially in the case of triple-negative cancer (HR = 0.18; 95% PI, 0.10–0.31) and HER2 positive (HR = 0.32; 95% PI, 0.21–0.47), as well as an increase in overall survival (HR = 0.22; 95% PI, 0.15–0.30) [86]. The results obtained were not dependent on subsequent adjuvant therapy. The pCR obtained through induction systemic therapy was considered a favorable prognostic factor for breast cancer [96].

## 5. Biological Types of Breast Cancer

Routinely determined elements of the pathomorphological examination seem insufficient to predict the clinical course of breast cancer, which makes it difficult to make appropriate therapeutic decisions. The diverse clinical course of cancers with similar morphological characteristics is due to their different gene profile.

The study of gene expression allowed the identification of five molecular subtypes of breast cancer, such as: luminal A, luminal B, HER-2 positive non-”luminal”, basal-like and special histological types. These surrogates correspond to the immunophenotypes of cancer cells determined according to pathological criteria. 

The luminal type A is characterized by high expression of genes associated with the activity of estrogen receptors and at the same time low expression of genes associated with proliferation and genes associated with expressed by the HER2 receptor.

Luminal type B is characterized by a positive ER status associated with low expression of genes associated with this receptor and higher than in type A expression of genes associated with the assessed proliferation by marking Ki-67. A panel of panelists in St. Gallen recognized the meltdown and expression of the Ki-67 as factors that could be used to differentiate between tumors of luminal type A and subtype B [97]. This is important in the prognostic assessment, which is better in type A. 

The next type is basal-like breast carcinoma, also called triple negative cancer due to the absence of estrogen and progesterone receptors and the lack of expression of the HER2 receptor- consequently, there is no expression of genes associated with these receptors. The group of patients with this type of cancer with metastases to the cerebellum is particularly interesting, in their case the use of biological markers (CK 5/6, HER1, c-KIT) can help in the restoration of the basal subtype similar and dissimilar, nevertheless, their clinical usefulness is ambiguous. 

The molecular subtype of breast cancer HER2- positive is characterized by overexpression of *HER2* combined with the absence of ER and PR.

Breast cancer is the most common cancer in women. Every year, the results of many clinical trials are published, only some of which cause a change in the standard of conduct. Treatment rules for patients with early breast cancer are updated every two years as part of a consensus set by experts St. Gallen International Breast Cancer Conference. Similarly, the European Society for Medical Oncology (ESMO) is developing its recommendations for the treatment of patients with breast cancer at an early stage. Recent Recommendations from St. Gallen (2019) highlight the progress that has been made, particularly in the management of HER2-positive and triple-negative breast cancers with residual disease after preoperative treatment [98]. 

## 6. Breast Cancer Treatment

The basic types of surgical procedures used in women treated for breast cancer are:-tumor excision;-mastectomy;-excision of the sentinel lymph node;-excision of the armpit lymphatic system.

Breast amputation is performed in patients who, due to the severity of the disease, do not qualify for breast-sparing treatment or do not agree to perform breast-sparing surgery. Breast amputation involves the removal of the entire breast and the entire skin covering the mammary gland (the exception is subcutaneous amputation). It is possible to make:-simple amputation—this is most often a palliative procedure in patients who are not eligible for radical treatment;-subcutaneous amputation, consisting in the removal of breast gland tissue and the nipple-areola complex, but leaving the skin;-modified radical mastectomy according to the Patey method, consisting in the removal of the mammary gland, lymph nodes of the axillary fossa, pectoral muscle minor and fascia of the pectoral muscle major;-modified radical mastectomy according to the Madden method, consisting in the removal of the mammary gland along with the fascia of the pectoral muscle major and the lymph nodes of the armpit in one tissue block;-radical mastectomy according to the Halsted method—performed in patients who have been diagnosed with infiltration of the cancer process on the pectoral muscles, consists in the removal of the mammary gland, lymph nodes of the axillary fossa, pectoral muscle larger. This treatment is currently rarely used [99,100].

Currently, breast conserving therapy (BCT), which is a method used in early forms of cancer, is becoming more and more widely used, and is characterized by the same effectiveness. Patients who meet the criteria for eligibility for sparing treatment, in accordance with the guidelines of the Association of Breast Surgery, should be offered the opportunity to choose between this treatment and mastectomy. Surgical treatment of breast cancer can take the form of sparing treatment consisting of:-removal of the tumor along with the margin of healthy tissues;-quadrantectomy;-surgery within the axillary fossa (all lymph nodes of the axillary fossa—axillary lymphadenectomy or sentinel lymph node).

The main purpose of surgery of patients treated for breast cancer is oncological completeness. Both in the case of radical mastectomy and breast conserving therapy, the appropriate cosmetic effect is important [98]. In the treatment of breast cancer, in addition to surgical intervention, adjuvant treatment is used consisting in the use of radiotherapy, chemotherapy, hormone therapy and immunotherapy or a combination of these methods. Radiation therapy is used in all patients treated with methods that spare the mammary gland, it reduces the risk of recurrence of the disease process. Indications for the use of adjuvant radiotherapy also include the occurrence of metastases in at least four axillary lymph nodes and the presence of positive tissue margins. The chest area and nodal fields are irradiated. Another type of adjuvant treatment is chemotherapy involving the use of cytostatics. This method is used in case of generalization of the disease process. It can be associated with radiation therapy. In breast amputees with indications for supplemental radiotherapy, it should be performed after the end of adjuvant chemotherapy. Hormone treatment is used in patients with breast cancer with estrogen receptor expression; it is used regardless of age and menopausal state. Another reason for using hormone therapy is to reduce the number of hormones secreted and alleviate the ailments and symptoms associated with cancer [101].

Thanks to advances in diagnostics, modern oncology can offer a personalized course of treatment—adapted to the characteristics of a given cancer. Doctors gained access to multigene tests, which, when used in a properly selected group of patients, are a valuable diagnostic tool. They help to plan the optimal treatment for a given patient and assess the likelihood of recurrence of the disease.

Diagnostic tests are used to measure the activity of a group of genes in breast cancer cells such as the Oncotype DX Breast Recurrence Score.

Its use was presented in the TAILORx clinical trial [102]. The study enrolled 10,273 breast cancer patients without lymph node metastases, estrogen receptor expression, and HER2 receptor expression. Based on the Oncotype DX test, patients with an intermediate risk of cancer recurrence were randomly assigned to an arm of the study in which only hormone therapy was used or to an arm in which patients were given chemotherapy together with hormone therapy. It was found that in the studied group of patients, independent hormone therapy is no less effective than combined chemotherapy with hormone therapy. Thanks to the results of a groundbreaking study, it is possible to safely avoid chemotherapy in the case of up to 70% of patients diagnosed with the most common form of breast cancer. The Oncotype DX Breast Recurrence Score is a diagnostic test which, based on the analysis of the expression of 21 genes, distinguishes three prognostic groups: with low, intermediate, and high risk of recurrence. The result of the study may help doctors make decisions about the optimal treatment of patients in the early stages of invasive breast cancer showing estrogen receptor activity (ER+), without expressing epidermal growth factor receptors (HER2-negative). The Oncotype DX test determines in this case the likelihood of a beneficial effect of the use of chemotherapy in combination with hormone therapy on treatment. Genetic tests in Oncotype DX work well in patients with Luminal A and B cancers, i.e., with positive estrogen and progestogen receptors, negative HER 2 and “clean” lymph nodes. Especially patients under 50 years of age are the group that can benefit most from the individualization of treatment. Patients with early breast cancer and good prognostic factors (ER+, PGR+, HER2−) are eligible for surgical treatment in the first place. After excision of the breast tumor and its histopathological evaluation and assessment of sentinel nodes (if there are no metastases in them), there is time to perform tests such as OncotypeDX or Mammaprint. The results of the tests make it easier to decide whether the patient should benefit from hormone therapy or chemotherapy will also be necessary and often put a dot over and in the qualification for treatment and allow the patient to be sure of the correctness of the choice of a specific therapy.

## 7. Recent Treatments for Triple Negative Breast Cancer

Among breast cancers, triple negative breast cancer (TNBC) is the most aggressive, and for its histochemical and molecular characteristics is also the one whose therapeutic opportunities are most limited. In case of breast cancer, a significant clinical problem is provided by the group of patients with no expression of any receptors, qualifying to hormonal therapy or target therapy against HER2 (the human epidermal growth factor receptor-2). The subtype of the disease, characterized by the lack of expression of estrogen receptors (ERs), the progesterone receptor (PR) and HER2, is referred to as the triple-negative breast cancer (TNBC). The triple-negative subtype constitutes approximately 15–20% of all breast cancer cases, its incidence being higher among younger women and is characterized by different biological features, unfavorable clinical course and poor prognosis. During the recent years, a thesis has been put forward that triple-negative breast cancer is a separate, heterogenic subtype of breast cancer, formed in the mechanism of different oncogenesis pathways, characterized by different prognoses and dependent on various clinical, pathological, and genetic factors. Despite its aggressive clinical course, the triple-negative breast cancer responds to chemical therapy, the response rate being very high. However, the disease recurrences are very frequent, while the lack of targeted therapy makes this cancer subtype very unfavorable from the prognostic point of view. No unequivocal principles of management have till now been proposed in the TNBC subgroup.

PARP inhibitors. So far, the Food and Drug Administration (FDA) has approved olaparib and thalasoparib for use in patients with advanced breast cancer with a germinal BRCA1/2 mutation [103,104,105]. The effectiveness of thalasoparib was proven in a phase III study, in which this drug was compared with standard chemotherapy, and its choice depended on the attending physician (in practice, capecitabine, vinorelbine, gemcitabine, eribulin). The I-row endpoint was progression-free survival (PFS). This study showed that thalasoparib was associated with a longer PFS duration (8.6 vs. 5.6 months, *p* < 0.001), thalasoparib was also better tolerated. Forty-five percent of patients were TNBC; 55% were HR+. Olaparib was validated based on a phase III study, with a very similar design to the EMBRACA study (with thalasoparib). The olaparib study showed that the use of this drug, compared to standard CHT, was associated with statistically significant PFS prolongation (7 vs. 4.2 months, *p* < 0.001). The tolerability of the treatment was also better. Patients eligible for the study, in addition to the current germline BRCA mutation, had to be HER2-. In both of the above studies, patients were previously treated (with anthracyclines and taxanes) and hormonotherapy [104,106].

Sacituzumab govitecan is a conjugate antibody directed against the surface tropoblast antigen Trop-2 along with the active metabolite irinotecan SN-38 [106]. A Phase 1/2 study evaluated the safety of sacituzumab govitecan in patients with advanced TNBC who had previously been treated with two chemotherapy (CHT) lines. Other endpoints included objective response rate, length of response, degree of clinical benefit, PFS, and OS. This study showed that the use of the above drug was associated with a benefit in the form of long-term objective responses. The 108 patients with triple-negative breast cancer had received a median of three previous therapies (range, 2 to 10). Four deaths occurred during treatment; three patients (2.8%) discontinued treatment because of adverse events. Grade 3 or 4 adverse events (in ≥10% of the patients) included anemia and neutropenia; 10 patients (9.3%) had febrile neutropenia. The response rate (3 complete and 33 partial responses) was 33.3% (95% confidence interval [CI], 24.6 to 43.1), and the median duration of response was 7.7 months (95% CI, 4.9 to 10.8); as assessed by independent central review, these values were 34.3% and 9.1 months, respectively. The clinical benefit rate was 45.4%. Median progression-free survival was 5.5 months (95% CI, 4.1 to 6.3), and overall survival was 13.0 months (95% CI, 11.2 to 13.7). The main adverse reaction was hematological toxicity. 

Immunotherapy as monotherapy. Documented efficacy in TNBC has a doublet of atezolizumab along with nab-paclitaxel-study IMpassion130 [107,108]. Anti-PD-1 or anti-PD-L1 monotherapy is still under investigation, but the U.S. FDA has approved the use of pembrolizumab in previously treated BC patients, after exhaustion of therapy options that have shown microsatellite instability or dMMR (The National Comprehensive Cancer Network (NCCN) recommendations). Recently, the results of meta-analyses and systematic reviews have appeared, which indicate the possible effectiveness of anti-PD1 and anti-PD-L1 antibodies in patients with TNBC. Studies have evaluated the efficacy of pembrolizumab, atezolizumab and avelumab, including their effects on ORR, PFS, OS. The results are promising, mainly in the group of patients expressing PD1 or PD-L1, especially when immunotherapy is used in the 1st line of treatment. It seems important to confirm the effectiveness of immunotherapy in TNBC characterized by a particularly poor prognosis, the treatment of which is currently limited to standard CHT.

In the aforementioned 2019 IMpassion130 study, it was shown that the use of atezolizumab with nab-paclitaxel in 1 line of treatment compared to nab-paclitaxel and placebo in patients with advanced TNBC was associated with an increase in the median PFS from 7.2 to 5.5 months (*p* = 0.002) in the ITT population, and in the PD-L1 expressing population 7.5 vs. 5.0 months (*p* < 0.001). However, in August 2021, the company producing atezolizumab voluntarily withdrew this drug from the indications for the treatment of TNBC. Currently, the only registered checkpoint inhibitor is pembrolizumab—for neoadjuvant treatment along with CHT TNBC with a high risk of relapse, with follow-up adjuvant therapy—based on the Keynote 522 study, and for the treatment of advanced TNBC with PDL1 expression (CPS > 10) based on the Keynote 355 study, as well as in patients with MSI-H and dMMR. In the latter indication, dostarlimab-gxly is also registered. Research is ongoing on other checkpoint inhibitors, including the previously described monotherapy [109,110]. 

The work of Spini et al. [111] provides an overview of all evidence regarding the reuse of old, licensed non-cancer drugs for the treatment of TNBC, ranging from preclinical evidence to current clinical trials.

Beta-blockers (BBs) appear to be promising drugs for reuse in the treatment of TNBC. While BB has been shown to be beneficial in the treatment of TNBC, metformin, a promising molecule in preclinical studies, has shown no efficacy in treating women with TNBC. Metformin does not improve survival outcomes in the female population with TNBC compared to women who do not use TNBC. It is worth noting that two studies are underway on the use of metformin in clinical trials in patients with TNBC.

Articles by Shiao et al. [112] and Williams et al. [113] showed conflicting results for aspirin. While the first study showed a significant improvement in survival in Grade II/III women through the use of aspirin, Williams et al. did not demonstrate this benefit in the breast cancer study population (women with operative I-III TNBC at stage).

Recently, one phase II study on omeprazole activity in patients with operative TNBC was presented at the ASCO meeting, regardless of baseline fatty acid synthase expression (FASN) [114]. In vitro, proton pump inhibitors inhibit FASN activity and induce apoptosis in breast cancer cell lines. In this study, 42 patients were given omeprazole in combination with anthracycline-taxane (AC-T) until surgery and a complete pathological response (pCR) was investigated. A positive FASN score decreased significantly for omeprazole from 0.53 (SD = 0.25) at baseline to 0.38 (SD = 0.30; *p* = 0.02), and the drug was well tolerated without known Grade 3 or 4 toxicity. In addition, the percentage of pCT was 71.4% (95% CI: 51.3–86.8) in patients with FASN+ and 71.8% (95% CI: 55.1–85.0) in all enrolled patients, indicating that omeprazole, in addition to neoadjuvant AC-T, provides a promising PCR rate without adding toxicity. From the literature obtained, BBs seemed to be more promising drugs for repurposing.

Agents that target angiogenesis have shown limited efficacy for human triple-negative breast cancer (TNBC) in clinical trials [113]. Considering the recommendations of the National Comprehensive Cancer Network (NCCN), the only drug that improved the endpoints of studies evaluating the effectiveness of anti-VEGF drugs with chemotherapy was bevacizumab. Ramucirumab, sorafenib and sunitinib were also studied [115]. 

According to NCCN:-study E2100 (more than 700 patients) evaluated the combination of bevacizumab with paclitaxel vs. paclitaxel with placebo in line 1 treatment for breast cancer recurrence/spread. This study showed that the addition of bevacizumab allowed for prolongation of the median PFS;-a similar study (more than 700 patients) evaluated the doublet bevacizumab plus docetaxel vs. docetaxel with placebo and in this study also achieved improvements in PFS in the combination group (AVADO study);-in the RIBBON-1 study, bevacizumab was attached to capecitabine, to taxane (docetaxel or nab-paclitaxel), to anthracyclines—here also PFS elongation was achieved by adding bevacizumab to CHT (study with 2nd line of treatment; the next study with 2nd line was IMELDA-elongation and PFS and OS were shown)-the last study mentioned by the NCCN was the Phase III CALGB 4050 study, which evaluated the addition of bevacizumab to nab-paclitaxel in line 1 of advanced TNBC treatment and achieved a median PFS of 7.4 months. In general, research shows that the addition of bevacizumab has an effect on ORR and PFS, but not OS and QoL.

One study showed that bevacizumab used in neoadjuvant lengthened both PFS and OS slightly (NSABP B-40 study). 

## 8. The Role of Non-Coding RNAs in Breast Cancer

The development of molecular biology has made it possible to conduct research at the level of the human genome. In 2003, its full sequence was published. Subsequently, it was discovered that only 1.2% of human genetic material encodes protein, with 93% of genes being transcribed. The huge pool of non-coding RNA molecules has aroused great interest among scientists. The subject of careful analysis in breast cancer became microRNAs, single-stranded RNA molecules with a length of 21 to 23 nucleotides, regulating the expression of other genes [116,117,118,119]. 

The first reports of the possible significance of altered miRNA expression in breast cancer were published in 2005. Over the past decade, several miRNA molecules involved in the initiation, progression, and metastasis of breast cancer have been identified [104]. The relationship between the expression of individual miRNAs and the clinical-pathological features of breast cancer, or the response to causal treatment of this malignant tumor,” has also been determined [115]. For example, studies have shown that in triple-negative breast cancer, there is an overexpression of oncogenic molecules miR-21, miR-210, miR-221, which is associated with a shorter disease-free time and worse survival [86]. Molecules with reduced expression, and therefore suppressor potential, are, for example, miR125-b in the case of HER-2 positive cancers, or miR-520 in hormone-dependent cancers [117,118].

Singh and Mo presented a review article on miRNA families, which play an important role in the course of the discussed malignant tumor. They paid attention to the miR-10 family, from which miR-10a and miR-10b are involved in the development and metastasis of breast cancer [116]. Overexpression of miR-10b is associated with a higher degree of cancer according to the TNM classification (larger size of the primary tumor, presence of metastases in the lymph nodes), a greater degree of cellular proliferation, overexpression, or amplification of the HER-2 receptor [117]. 

However, it is negatively correlated with the presence of steroid receptors and the concentration of E-cadherin, which seems to play a role in the suppression of the metastasis process in the EMT (Epithelial-mesenchymal transition) mechanism [105]. Metastasis, as well as a worse course of particularly ductal breast cancer and consequently shorter overall survival time is also associated with the oncogenic miR-21 family [104]. Among the families of suppressor miRNAs, with reduced expression in cancerous breast tissue compared to healthy tissue, the aforementioned authors mentioned the miR-200 family and miR-205 and miR-145. 

MiR-200 and miR-205 probably inhibit the metastasis process associated with the EMT mechanism, and miR-145 affects cell apoptosis [119].

On the other hand, in a 2019 review article by Loh et al., the decisive oncogenic potential of the miR-200 family was described [120]. Increased concentrations of individual miR-200s were associated not only with breast cancer’s ability to form distant metastases, but also with resistance to chemotherapy [121].

The relationship between the expression of the rich miRNA family and the cell cycle, including the disturbed tumor cell cycle, certainly requires further analysis. However, there is no doubt that the molecules in question have a huge prognostic, predictive and therapeutic potential. Promising research results have prompted scientists to search for new regulatory molecules. 

Of particular recent interest are long non-coding RNAs, or lncRNAs. An extensive description of lncRNA was presented by the authors of this article in a review paper Smolarz et al. [122].

## 9. Summary

The assessment of the achievements over recent years in the treatment of patients with breast cancer, with the simultaneous lack of fully satisfactory results and satisfactory solutions, suggests that further progress in the development of new methods of combating cancer will bring us closer to a new era in this field.

## Figures and Tables

**Table 1 cancers-14-02569-t001:** Risk factors for breast cancer [8].

Hormonal and reproductive	Early age of the first menstruation
	Late age of the last menstruation
	The first reported pregnancy at a late age (after 30 years of age)
	No pregnancies
	Postmenopausal condition
	Use of oral contraception
	Use of hormone replacement therapy
Related to physiological factors and health status	Older age (increased risk from 35 years of age)
	Family history of breast cancer
	Breast, ovarian and endometrial cancer in the past
	Occurrence of benign changes in the breasts, proceeding with the presence of atypical hyperplasia
	Ionizing radiation, used in connection with, for example, Hodgkin lymphoma therapy
	Rapid growth in adolescence and high growth in adulthood
	Infection with an oncogenic virus (e.g., Epstein–Barr)
Nutritional	Western type diet
	Excessive consumption of fats, especially animal fats
	High consumption of red and fried meat
	High iron intake
	Development of overweight/obesity after menopause
	Low consumption of fresh vegetables and fruits
	Low intake of phytoestrogens (isoflavones, lignans)
Other lifestyle-related	Regular moderate/high alcohol consumption
	Lack of regular physical activity
	Night work

**Table 2 cancers-14-02569-t002:** Epithelial precursor lesions and invasive lesions of the mammary gland [74].

Epithelial Precursor Lesions	Invasive Changes
Atypical lobular hyperplasia	Nonspecific weaving cancer (NST) Medullary carcinoma
Lobular carcinoma in situ	Oncocytic carcinoma
Ordinary wired hyperplasia	Cancer with rich fat weaving
Cylindrical cell changes	Cancer with rich glycogen weaving
Atypical ductal hyperplasia	Sebaceous cancer
In situ ductal carcinoma	Microinvasive cancer
	Lobular cancer
	Tubular cancer
	Sit-like cancer
	Mucous cancer
	Cystadenocarcinoma
	Invasive micro beard carcinoma
	Cancer with apocrine differentiation
	Metaplastic cancer
	Rare cancers and types of salivary gland cancers

**Table 3 cancers-14-02569-t003:** Assessment of the degree of histological malignancy [75].

Feature	Score (Points)
Formation of coils and glands	
>75%	1
10–75%	2
<10%	3
Nuclear pleomorphism (degree of nuclei atypia)	
Small, regular, homogeneous	1
Moderately enlarged and heterogeneous	2
Clearly pleomorphic	3
Number of figures of cancer cell division	
Depends on the size of the microscope’s field of view	From 1 to 3
The degree of histological malignancy after summing up the above results	
Grade 1	3–5
Grade 2	6–7
Grade 3	8–9

**Table 4 cancers-14-02569-t004:** VIII edition of the pTNM classification.

pT		
TX		It is impossible to evaluate the tumor
T0		Tumor absent
Tis		Cancer in situ
	Tis (DCIS)	Ductal carcinoma in situ
	Tis (Paget)	Paget’s cancer (no infiltrating or in situ cancer in the breast)
T1		Infiltrating cancer ≤ 20 mm
	T1mi	Micro-infiltrating cancer ≤ 1 mm
	T1a	Infiltrating cancer > 1 mm i ≤ 5 mm
	T1b	Infiltrating cancer > 5 mm i ≤ 10 mm
	T1c	Infiltrating cancer > 10 mm i ≤ 20 mm
T2		Infiltrating cancer > 20 mm i ≤ 50 mm
T3		Infiltrating cancer > 50 mm
T4		Infiltrating cancer of any size with invasion of the chest wall and skin (ulcer or satellite nodules)
	T4a	Infiltration of the chest wall (but not the pectoral muscles)
	T4b	Ulcer, satellite nodules, swelling of the skin that does not meet the criteria for inflammatory cancer
	T4c	T4a + T4b
	T4d	Inflammatory cancer
pN		
NX		Unable to evaluate nodes
N0		There are no metastases to regional lymph nodes
	N0 (i-)	There are no metastases to regional lymph nodes in the HE and IHC study
	N0 (i+)	Isolated cancer cells (HE or IHC) ≤ 0.2 mm or < 200 cells were detected
	N0 (mol-)	There are no metastases to regional lymph nodes (also molecular biology techniques)
	N0 (mol+)	Molecularly detected metastatic features with negative HE and IHC image
N1		Metastases in 1–3 regional lymph nodes
	N1mi	Micrometastases > 0.2 mm or > 200 cells in 1–3 lymph nodes
	N1a	Metastases in 1–3 regional lymph nodes (including at least one >2 mm)
	N1b	Metastases (or micrometastases) in the internal thoracic lymph nodes (SLNB)
	N1c	N1a + N1b
N2		Metastases in 4–9 regional lymph nodes
	N2a	Metastases in 4–9 regional lymph nodes (including at least one >2 mm)
	N2b	Metastases (or micrometastases) in the internal thoracic lymph nodes in the absence of metastases in the axillary lymph nodes
N3		Metastases in the ≥10 regional lymph nodes or in the supraclavicular node or >3 axillary and thoracic
N3a		Metastases in the ≥ 10 regional lymph nodes (axillary) or in the subclavian node (third floor of the axillary fossa)
N3b		Axillary > 3 and thoracic internal
N3c		Metastasis in the supraclavicular node
pM		
M0		No metastases
M0 (i+)		Cancer cells detected microscopically or by molecular biology techniques in blood or other tissues, excluding regional lymph nodes ≤ 0.2 mm (or ≤200 cells), in the absence of other signs of metastasis
M1		Metastases to distant organs (clinically or pathologically)

**Table 5 cancers-14-02569-t005:** HER-2 receptor IHC rating scale, interpretation.

Result	Interpretation
0—no reaction or color reaction in the <10% of infiltrating cancer cells	Negative state
1+—discontinuous coloration, complete membrane staining in the <10% of infiltrating cancer cells	Negative state
2+—weak or medium complete membrane staining in >/= 10% of infiltrating cancer cells	Ambiguous (borderline) state, requires in situ hybridization of the same material or reassessment of IHC or ISH from other material of the examined tumor
3+—Strong complete membrane staining in >/= 30% of infiltrating cancer cells	Positive state

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
