# Peer review of "Breast Cancer—Epidemiology, Classification, Pathogenesis and Treatment (Review of Literature)"

_cancers, 2022, doi:10.3390/cancers14102569_

Round 1

Reviewer 1 Report

Well done and fairly comprehensive review of breast cancer. Certain aspects related to treatment (in particular surgical treatment) are missing as well as some diagnostic and prognostic aspects (i.e. oncotyping).

Author Response

Thank you for your review.

I would like to kindly ask you to reconsider the publication of our revised paper:

" Breast cancer - epidemiology, classification, pathogenesis and treatment (review of literature)”.

I hereby provide responses to the reviewers and list the changes that have been made in the revised version of our paper.

We have added chapter 6. Breast cancer treatment, in which we included the aspect of treatment (in particular surgical treatment) and oncotyping.

I hope you find our revised Manuscript satisfying so that it can meet the criteria of publication in your Journal.

Looking forward to hearing from you,

Yours sincerely,

Beata Smolarz

Reviewer 2 Report

Dear authors,

Your review provide an overview of the risk factors, classification and treatment of breast cancer. I have some comments / suggestion:

  1. Line 58: here you report the incidence in Poland. I suggest to integrate with the incidence at line 38-46 to be more consistent.
  2. Line 62. You state that morbidity in adult women of premenopausal age has doubled. Some reasons why? I suggest you deepen this point.
  3. Line 63: you reported data on mortality while the paragraph where you report this result was at line 47-57. Please be consistent.
  4. Line 84: delete “in men”
  5. Line 87: I suggest authors not to use “our country” but Poland instead. Moreover, i suggest to have a more broad international prospective.
  6. Line 116 please check the font and the editing
  7. Line 119: please put inside the brackets the acronyms and outside the full name description
  8. Line 130 I think a reference is missing
  9. Please check the indent paragraphs all over the manuscript and be sure that you start a new line only when the topic before is concluded (e.g., line 330). This could be misleading for readers.
  10. Line 174: “Nulliparous women and those who became pregnant for the first time after the age 174 of 30. have an increased risk of getting sick 2 – 5 times. “ 2-5 times more?
  11. I suggest the authors to report all risk factors in a specific table. You can add also the studies you fund reporting the risk factors and perhaps how much the single factor affects the development of breast cancer. Moreover, there are some differences in developing specific type of breast cancer for each risk factor?
  12. Chapter 2.6: There are correlations also on those breast cancer on the basis of hormonal status?
  13. Line 248: please provide reference.
  14. Line 256: Could benign changes also be explained by exposure of other risk factor for breast cancer?
  15. Please check punctuation all over the manuscript. Sometimes dots are  missing.
  16. Chapter 6. Some reasons why you focus only on TNBC? It is not clear from the manuscript and from the title which it seems referred to all breast cancers.
  17. Line 622: Please provide some results.
  18. Line 635: What did you mean by “in the case of breast cancer”?
  19. Given the few therapeutic options for TNBC, as suggested by other reviews (PMID: 32728387, PMID: 31607128), other approaches such as repurposing of drugs could be a valuable option to look at. I suggest you to comment more on this topic.
  20. What about anti-VEGF drugs? Are they used to treat TNBC? Some evidence in combination with the therapies you cited? I believe that these drugs could be used as chemosensitizers.
  21. Chapter 7. Please provide a more comprehensive summary of the evidence you retrieved.

Author Response

Thank you for your review.

I would like to kindly ask you to reconsider the publication of our revised paper:

" Breast cancer - epidemiology, classification, pathogenesis and treatment (review of literature)”.

I hereby provide responses to the reviewers and list the changes that have been made in the revised version of our paper.

We are submitting an article that we have tried to rebuild in terms of the reviews we have received. We tried to clarify in the text and include answers to all the questions of the reviewers. We hope that we have improved the article as suggested by the reviewers.

I hope you find our revised Manuscript satisfying so that it can meet the criteria of publication in your Journal.

Looking forward to hearing from you,

Yours sincerely,

Beata Smolarz

Reviewer 3 Report

Reviewer

Comments and Suggestions for Authors

This study evaluates the epidemiology, classification, pathogenesis, and treatment of breast cancer. The study collates relevant research well but does not meet the full review requirements.

A review should answer a single, clear, focused research question. This manuscript does not present a clear question in the Abstract. The closest is the aim written in the Abstract: “The article presents a review of the literature on breast carcinoma - a disease affecting women in the world.”.

It is nice to have the relevant studies collated and discussed all in one place, but there is not much additional benefit. 

The manuscript requires major revisions to meet the criteria of a review.

Abstract:

  • The Abstract could be improved by including a description of the study methods and outcomes.
  • Specifically, the main goals of this study.
  • Aims not clear because, for example, the treatment section only discusses the form of triple-negative breast cancer 

Epidemiology:

  • Line 31 - The greatest value of this study is observed in the group of women aged 50-69 years. Please include the reference
  • Lines 32 -33 please insert references
  • Line 80 includes some repetition - Only 1% of cases of this malignant tumor affect men. The incidence of breast cancer in men is small.
  • Lines 28-29 have the same idea as in line 22
  • Line 62 Please rephrase this sentence “morbidity in adult women of premenopausal age (20–49 years)” 

Perimenopause can begin in some women in their 30s, but it often starts in women ages 40 to 44.

  • The specific research question is not clear.
  • No study design.

Body Text:

It would have been interesting to stratify the risk factors according to their importance.

Some risk factors are accompanied by a description of the pathophysiological mechanism, while other risk factors are not.

Section 2.1 – these are some suggestions

-    the 5-year survival rate for male patients was lower than for female patients (82.8% vs. 88.5%).

-    the risk of death in men was 43% greater than that in women during the follow-up period.

-    breast cancers in men - most are ER-positive, some are HER2 positive, and few have invasive lobular subtypes DOI: 10.1016/j.clbc.2018.06.013; tends to be ductal type and estrogen receptor and progesterone receptor-positive https://doi.org/10.1038/s41598-021-91131-4

Section 2.2 

  • “50% of these patients are in the age range of 50-69 years [8].” The global increase in BC incidence is seen in all age groups and is highest in women under 50 years DOI: https://doi.org/10.1016/j.eclinm.2021.100985

Section 2.3 

  • countries with a very high human development index (HDI) had the highest premenopausal and postmenopausal breast cancer incidence, whereas countries with low and medium HDI had the highest premenopausal and postmenopausal mortality, respectively. DOI: 1016/S2214-109X(20)30215-1

Section 2.4

  • Lines 152-154 please mention those studies
  • Breast cancer risk in transgender people receiving hormone treatment is another problem.

doi: 10.1136/bmj.l1652

  • It can be noted an increased risk of breast cancer in trans women compared with cisgender men and lower risk in trans men compared with cisgender women.
  • In trans women, the risk of breast cancer increases during a relatively short duration of hormone treatment. 
  • Please add newer references 1136/bmj.m3873
  • current/recent HRT exposure may contribute to the incidence of luminal breast cancer tumors in particular. DOI: 2147/IJWH.S311696

Section 2.6

  • Please add newer references 1056/NEJMoa1913948, 10.1016/j.breast.2021.08.011
  • It is believed that <10% of breast cancers are genetically determined. 
  • Line 248 same as line 219

Section 2.8 

  • Please specify the mechanism in case of other risk factors
  • IR may increase the risk of ER - breast cancer in particular, although some studies report no difference between the risk of ER+ and ER− cancers. 1007/s00204-020-02752-z

Section 2.10 

-    nutrition influences the prognosis of breast cancer. Nevertheless, the level of evidence of the results was insufficient to make recommendations. Ultimately, a healthy and balanced diet could be encouraged to reduce global mortality. 10.1016/j.bulcan.2019.08.009

-    nutritional intervention in BC patients may be considered an integral part of the multimodal therapeutic approach. However, further research utilizing dietary interventions in large clinical trials is required to definitively establish effective interventions in these patients, to improve long-term survival and quality of life. 10.3390/nu11071514

-    nutritional interventions represent crucial strategies in the field of Public Health. Nutrigenomics and nutriproteomics arise from integrating nutritional, genomics, and proteomics specialties in the era of postgenomics medicine. 10.3390/nu12020512

Section 2.11 

There are a few additions to be made:

-    Obesity is a recognized risk factor for breast cancer and recurrence development even when patients are treated appropriately. 

-    Obese women are less likely to undergo breast reconstruction than normal-weight women, and those who have surgery experience more surgical complications. 

-    systemic chemotherapy is less effective,

-    endocrine therapy is less effective

-    obese women are at increased risk for local recurrence than normal-weight women.

-    efficacy of cancer treatments is significantly lower in obese breast cancer survivors. 

Section 2.12

There are a few recommendations to be made:

-    Nicotine promotes breast cancer metastasis by stimulating N2 neutrophils and generating a pre-metastatic niche in the lung. 10.1038/s41467-020-20733-9

-    Chemoresistance effects of nicotine were demonstrated in these cells. These findings demonstrated the harmful effects of nicotine following metastasis of cancer, owing to the chemoresistance produced through uninterrupted smoking, which may impact the effectiveness of treatment. 10.3892/etm.2018.6149

  • I recommend introducing another risk factor - Section 2.13 “chemical environment and pollution”
  • effect on the signaling pathways involved in metastatic tumor cells' emergence and progression.

Section 4 

  • please include lymph node biopsy

Line 395 - core needle biopsy instead of coarse needle biopsy

  • Please specify the main breast cancer biomarkers:

Line 420 

-    Nottingham Histologic Score system, also termed “Bloom-Richardson-Scarff classification in Elston-Ellis modification”

-    At present we have a system with diagnostic categories of carcinomas that are still based on morphologies with possible overlaps (histological types).

-    A discussion on the influence of treatment based on histological grade or biomarker expression would be interesting

Line 484 

  • the gold standard – Neoadjuvant endocrine therapy represents a feasible and effective treatment option, especially in ER+ HER2 negative postmenopausal patients, with AI administered preoperatively for 3–6 months.

Lines 492-507 and lines 509-519 

  • Please insert references

Section 4.4  

Although rare, HER2 mutations appear as important molecular changes that need to be identified, for example, in patients with metastasis, tumors with HER2 mutations may respond to specific tyrosine kinase inhibitors. HER2 mutation may also be a mechanism of resistance to anti-HER2 therapeutic compounds. 

Line 536 

  • Please rewrite the following proposition “The percentage of colored testicles of cancer cells is the value of the cell proliferation index Ki67.”

Section 6 deals with recent treatment in triple-negative breast cancer, although the title specifies the generic term for treatment. Please clarify this aspect.

Line 607 - thalasoparib instead of talazoparib

Lines 608-610 

  • Please rewrite the following phrase “This study showed that thalasoparib was associated with a longer PFS duration (8.6 vs. 5.6 months, p<0.001), talazoparib was also better tolerated.”

Line 635 

  • Specify the reference

Please insert the limitations section

Kind regards

Author Response

(The authors gave the same response as above.)

Reviewer 4 Report

the work has strengths and weaknesses. among the former, there is the effort of an exhaustive re-examination of the literature. however the main limitations are represented by actually an all-encompassing zibaldone that inevitably loses its organicity and jumps from risk factors to non-coding mRNA sequences without any logic.

moreover, being a review, the references shoulb be particularly well managed while there are so many mistakes, ie

line 31: the greatest value of this study. what study???

36: better "plus" instead of and

2.6: the chapter is too generic

257: the right ref number is 33

343: recent studies are 2013 and 2016... not so recent

395 core needle, not coarse

399: repetition

431:according to quality indicators, more than 10 nodes

469 repetition

543 ref 85 instead of 83

556: spring... I can't see the reference

564: ref85 is wrong

576: associated with... what?

579: panel of panelist...

586-589: delete

592-600: out of context

607: talazoparib

615delete bracket after taxanes and put after therapy

664: has also been ... what

Author Response

(The authors gave the same response as above.)

Round 2

Reviewer 2 Report

Dear authors, thank you for your answers. I believe that your article is now enhanced. 

I have really few other comments: 

  1. Line 27: Please not that breast cancer does not only affect women.
  2. Line 70: What do you mean with "we still pale"? Please also use a more broad prospective

Reviewer 3 Report

Dear Authors, 

Despite the fact that this review is very long, your work in trying to systematize such a vast subject is to be appreciated.

Kindest regards